

# TOAST 1.0: Tropospheric Ozone Attribution of Sources with Tagging for CESM 1.2.2

Tim Butler[1], Aurelia Lupascu[1], Jane Coates[1], and Shuai Zhu[1,2]

[1]Institute for Advanced Sustainability Studies, Potsdam, Germany
[2]now at IBM Research - China

*Correspondence to:* Tim Butler tim.butler@iass-potsdam.de

**Abstract.** A system for source attribution of tropospheric ozone produced from both $NO_x$ and VOC precursors is described, along with its implementation in the Community Earth System Model (CESM) version 1.2.2 using CAM4. The user can specify an arbitrary number of tag identities for each $NO_x$ or VOC species in the model, and the tagging system rewrites the model chemical mechanism and source code to incorporate tagged tracers and reactions representing these tagged species, as well as ozone produced in the stratosphere. If the user supplies emission files for the corresponding tagged tracers, the model will produce tagged ozone tracers which represent the contribution of each of the tag identities to the modelled total tropospheric ozone. Our tagged tracers preserve $O_x$. The size of the tagged chemical mechanism scales linearly with the number of specified tag identities. Separate simulations are required for $NO_x$ and VOC tagging, which avoids the sharing of tag identities between $NO_x$ and VOC species. Results are presented and evaluated for both $NO_x$ and VOC source attribution. We show that northern hemispheric surface ozone is dominated year-round by anthropogenic emissions of $NO_x$, but that the mix of corresponding VOC precursors changes over the course of the year; anthropogenic VOC emissions contribute significantly to surface ozone in winter-spring, while biogenic VOC are more important in summer. The system described here can provide important diagnostic information about modelled ozone production, and could be used to construct source-receptor relationships for tropospheric ozone.

## 1 Introduction

Tropospheric ozone is an important air pollutant, as well as an contributor to anthropogenic radiative forcing of the climate (Monks et al., 2015). Major sources of ozone in the troposphere are transport from the stratosphere, and photochemical production involving reactions of oxides of nitrogen (NO and $NO_2$, collectively $NO_x$) and Volatile Organic Compounds (VOC), including methane. Almost all of this photochemical production is related to the conversion of NO to $NO_2$ by reaction with a peroxy radical produced during the oxidation of VOC (Atkinson, 2000). Due to its long lifetime in the troposphere (several weeks), ozone can be transported over intercontinental distances. Concentrations of ozone observed at any given location can be due to both transported ozone from elsewhere, and ozone produced from precursors emitted nearby.

Global chemistry-climate models are important tools for understanding the complex processes of chemistry and transport which affect tropospheric ozone, simulating its evolution and distribution under future climate change, and projecting how





this may change in response to precursor emission controls. Based on a suite of model simulations from the ACCMIP model intercomparison project (Lamarque et al., 2013), Young et al. (2013) found that while the ensemble average of modelled ozone mixing ratios generally agreed with the present-day distribution of tropospheric ozone well, the individual models showed large differences from each other. Furthermore, the models generally agreed on the sign of the difference between present-
day and both pre-industrial and future (late 2100s) conditions, but they tended to disagree strongly on the magnitude of these changes. The current state-of-the-art models show differing sensitivities in tropospheric ozone to changes in both climate and precursor emissions. In a more detailed comparison of the ACCMIP models with observed datasets, Parrish et al. (2014) showed that the models are not able to simulate the observed long term changes in tropospheric ozone, and concluded that more work is needed to improve the representation of chemistry and transport processes in models, as well as our understanding of
historical emission changes, before the models could be reliably used to simulate future changes in tropospheric ozone. Young et al. (2013) identified the need for improved diagnostic information about modelled ozone budgets in order to understand the differences between models.

Intercontinental source-receptor relationships for tropospheric ozone have been modelled in the HTAP (Hemispheric Transport of Air Pollution) project using a "perturbation" methodology in which emissions of ozone precursors in source regions
are reduced by some fraction (eg. 20%), and the resulting modelled ozone concentrations in receptor regions are compared with a base simulation in which the emissions were not perturbed (Fiore et al., 2009). An alternative approach for determining source-receptor relationships in model runs is a technique known as "tagging", in which ozone molecules are labelled with the identity of their source, allowing direct attribution of ozone concentrations to these sources in receptor regions (eg. Wang et al., 1998; Dunker et al., 2002; Sudo and Akimoto, 2007; Grewe et al., 2010; Emmons et al., 2012; Derwent et al., 2015;
Kwok et al., 2015; Grewe et al., 2017; Guo et al., 2017). Tagging (source apportionment) methodologies are complementary to perturbation (sensitivity) methodologies (Emmons et al., 2012; Grewe et al., 2017; Clappier et al., 2017). Since tagging methods can deliver detailed information about the provenance of modelled ozone concentrations, they could potentially be a useful tool for understanding the differences between models.

In this manuscript we describe and characterise a novel method for tagged source attribution of tropospheric ozone, and
contrast our approach with previous work. We present a review of prior tagging approaches in Section 2, then describe the implementation of our method in CESM 1.2.2 with CAM4 (Tilmes et al., 2015; Lamarque et al., 2012) in Section 3. The design of our model evaluation experiments is described in Section 4, and we evaluate and compare the results for both $NO_x$- and VOC-tagging in Section 5. Conclusions and outlook are presented in Section 6.

## 2   Review of tagging methods

There are many different examples of several different approaches to ozone tagging in both regional and global models. In this study, we focus on the attribution of ozone production to emitted precursors. Studies such as Wang et al. (1998), Sudo and Akimoto (2007) and Derwent et al. (2015) each tag ozone molecules based on the geographical model domains in which the





ozone molecules are formed, so do not directly attribute chemical ozone production to emissions of particular precursors, and will not be discussed further here.

Attribution of ozone production to emissions in models of atmospheric chemistry involves several design decisions and associated trade-offs:

– Is ozone production attributed to emissions of $NO_x$, VOC, or both? And if both, how is the chemical regime ($NO_x$- or VOC-limited) accounted for?

     – Is ozone production attributed explicitly for each chemical reaction producing ozone, or is the total instantaneous ozone production in each grid cell attributed according to the proportion of each precursor present?

     – Are tagged precursor species simulated explicitly, or are they grouped into chemical "families"?

– How does the tagging system treat the $O_3$-$NO_x$ null chemical cycle?

Since both $NO_x$ and VOC are involved in the chemical production of ozone, most tagging schemes attempt to attribute ozone production to both of these types of precursors. Two approaches for simultaneous attribution of ozone to both $NO_x$ and VOC have been used: determination of the chemical regime with attribution to the limiting precursor (either $NO_x$ or VOC); and equal attribution to both $NO_x$ and VOC precursors. In each case, additional tracers are added to the model, which track the emissions

of $NO_x$ and VOC species, which are typically labelled with the identities of their source sectors (eg. transport, industry, etc...) or source regions (eg. East Asia, North America, etc...).

Determination of the chemical regime is typically made according to the indicator ratio $PH_2O_2/PHNO_3$ (the ratio between the production rates of hydrogen peroxide and nitric acid). According to (Sillman, 1995), the chemical regime is $NO_x$- or VOC-limited if the ratio is above or below 0.35 respectively. This approach somewhat simplifies the highly complex chemistry

of ozone production, in which there is a transition regime of sensitivity to both $NO_x$ and VOC emissions. This approach is also typically used in regional modelling studies, where model grid cells are relatively small (compared with global models). VOC-limited chemical regimes are typically found in regions of very high $NO_x$ emissions, such as urban areas, which are not well-resolved by global models. Dunker et al. (2002) and Kwok et al. (2015) describe the use of this technique in the regional models CAMx and CMAQ respectively. In both cases, the tagging scheme determines whether ozone production in

each model grid cell is in a $NO_x$-limited or a VOC-limited chemical regime, and attributes all instantaneous ozone production to the limiting precursor, with tagged ozone tracers added in proportion to the relative concentrations of the tagged precursor tracers present in that grid cell. Tagged ozone tracers are chemically destroyed according to the modelled instantaneous ozone chemical loss rate. Such tagging schemes account for the rapid null cycles involving $O_x$ species by not considering their cycling reactions as part of the instantaneous ozone production or loss rates.

We are not aware of any global modelling study which has attempted to attribute ozone production to $NO_x$ or VOC precursors based on the chemical regime in each grid cell. Instead, ozone tagging at the global scale has been done either by focusing on only $NO_x$ precursors (eg. Emmons et al., 2012), or by giving equal weight to both $NO_x$ and VOC precursors (eg. Grewe et al., 2010, 2017; Guo et al., 2017). In each case, the production rate of tagged ozone tracers is determined explicitly with respect





to the rates of the underlying chemical reactions producing ozone, rather than the bulk instantaneous ozone production rate. Grewe et al. (2010, 2017) use rate constants from the base chemical mechanism as well as the full set of concentrations of the tagged species to explicitly calculate the production rate of each tagged species considering all of the possible combinations between differently tagged precursor reactants. In their scheme, tagged ozone is produced from reactions between tagged

NO and tagged peroxy radicals, and the ultimately produced ozone molecules inherit their tag identities from both types of precursors. Emmons et al. (2012) and Guo et al. (2017) take a different approach, and add extra reactions to the base chemical mechanism representing the transformations of the tagged precursors and the production of tagged ozone, relying instead on the chemical solver of their model to calculate the production and loss rates of tagged species.

Similarly to Grewe et al. (2010, 2017), Guo et al. (2017) also takes a combinatorial approach to the simultaneous attribution

of tagged ozone to both $NO_x$- and VOC-tagged precursors. They avoid the chemical mechanism becoming too large by only considering two tag identities ("East Asia" (EA) and "everywhere else" (EE)). Each reaction between a peroxy radical and NO then requires four corresponding tagged reactions: EA+EA, EA+EE, EE+EA, and EE+EE. The size of their tagged mechanism thus increases quadratically with the number of tag identities. In the case of the cross-reactions (EA+EE and EE+EA), the $NO_2$ produced from the reaction between NO and a peroxy radical is split into equal parts $NO_2$ from EA and $NO_2$ from EE, despite

the fact that the NO reactant in any given reaction can only have come from one of these regions. By using such a combinatorial approach, Grewe et al. (2010, 2017) and Guo et al. (2017) allow the transfer of tag identities between $NO_x$ and VOC species, which can produce tagged tracer concentrations which have no physical meaning. For example, Figure 5(b) of Grewe et al. (2017) attributes approximately 10 Tg of CO production per year to lightning, despite the fact that lightning is only a source of $NO_x$ in their model. Such an unphysical result could be obtained in their tagging scheme after decomposition of a molecule

of PAN (peroxy acetyl nitrate, an organic nitrate) which had been tagged as coming from $NO_x$ due to lightning. A similarly unphysical transfer of tag identity would be obtained in the approach used by Guo et al. (2017) if lightning were chosen as one of their tag identities.

The treatment of the $NO_x$-$O_3$ chemical cycle is another area in which ozone tagging schemes can produce unphysical results. As pointed out by Kwok et al. (2015), the approach of Emmons et al. (2012) treats the reaction between NO and $O_3$ (forming

$NO_2$) as chemical destruction of $O_3$. The subsequent rapid re-formation of $O_3$ from $NO_2$ photolysis is treated as new ozone production due to an emitted $NO_x$ precursor, effectively "overwriting" the identity of tagged ozone from remote sources with the identity of tagged $NO_x$ emissions from more nearby sources. The work of Grewe et al. (2017) does not suffer from this problem, because ozone is included in a chemical family ($O_x$ = odd oxygen = $O_3$ + O + $NO_2$ + others) which is preserved during fast chemical exchanges. Guo et al. (2017) do not give enough information to determine whether their approach also

suffers from this tag-overwriting problem.

While the use of the $O_x$ chemical family is essential to preserve the correct identity of tagged ozone species, the use of other chemical families for ozone precursors can introduce additional problems with tagging schemes. For example, Grewe et al. (2017) do not explicitly follow the propagation of tags through the full set of VOC oxidation intermediates, but instead only tag a single "NMHC" chemical family, which includes all VOC oxidation intermediate species, including the oxidation products

of methane, but excludes PAN. Their use of this NMHC family leads to the unphysical result from their Figure 5(d), in which





formation of PAN has been partially attributed to methane. There is no known chemical pathway in the atmosphere capable of transforming methane into PAN. This is not an inherent weakness of their tagging approach, but rather results from their choice of one chemical family to represent all VOC ozone precursors. In order to avoid such unphysical results, the choice of chemical families must be made carefully. Ideally, each individual VOC oxidation intermediate should be explicitly tagged.

Butler et al. (2011) introduced a method for recursively tagging all reactions involving VOC species in a chemical box model. They followed and tagged the oxidation pathways of all VOC intermediate products until they were fully oxidised, and thus no longer included in the chemical mechanism. Butler et al. (2011) used this method to determine the time-dependent ozone production potential of all VOC species in the MCM (Master Chemical Mechanism Saunders et al., 2003) by tagging each of the "primary" (emitted) VOC with its own identity, and were thus able to attribute ozone production from intermediate VOC
species back to the emissions of each primary VOC species, thus avoiding the use of a generic VOC chemical family. Butler et al. (2011) showed that the chemistry of VOC intermediate products can contribute significantly to the total ozone production from VOC over the timescales of several days after emission. Using this approach, it was feasible to tag each primary VOC and all of its intermediate oxidation products in the MCM with a unique tag, due to the way in which the interactions between different organic peroxy radicals are treated in the MCM. The peroxy-peroxy chemistry of each individual peroxy radical in
the MCM is represented as a unimolecular decay reaction with a rate constant proportional to the total concentration of all other peroxy radicals. As also noted by Ying and Krishnan (2010), if these peroxy-peroxy reactions are treated explicitly in a tagged chemical mechanism, the size of the tagged mechanism would scale quadratically with the number of tags, which would rapidly become too large for practical use. The technique of Butler et al. (2011) was subsequently applied for comparison of several VOC oxidation mechanisms by Coates and Butler (2015). In order to avoid the quadratic scaling problem, the chemistry
of the organic peroxy radicals in each chemical mechanism was rewritten in the MCM style, allowing the size of the tagged chemical mechanism to scale linearly with the number of tag identities.

In this manuscript we describe an extension to the ozone tagging system first described fully by Emmons et al. (2012). This extended tagging system improves upon the earlier work of Emmons et al. (2012), avoiding the various problems with previous tagging schemes described above.

– Our tagging scheme allows an arbitrary number of user-defined tag names in a single model run, with the size of the chemical mechanism increasing linearly with the number of tag identities.

– Our tagging scheme introduces new tagged tracers for members of the $O_x$ chemical family (which avoids the problem that ozone tags are destroyed by the null cycle involving $NO_x$).

– Our tagging scheme incorporates the recursive VOC tagging system of Butler et al. (2011), explicitly tagging each
intermediate VOC and avoiding the use of precursor families.

– Our tagging scheme avoids the possibility of VOC species being tagged with identities of $NO_x$ species (and vice-versa) by requiring that two separate model runs be performed, one with $NO_x$ tagging, and another with VOC tagging. The tagged $O_x$ produced during the conversion of NO to $NO_2$ can only be assigned to the tagged identity of the NO precursor, or the



tagged identity of the peroxy radical involved in each such transformation, depending on whether NO$_x$- or VOC-tagging is being used.

The extended tagging system allows a completely closed source attribution of tropospheric ozone to all precursors to be performed in two model runs, one with NO$_x$ tagging, and another with VOC tagging.

## 3 Implementation of NO$_x$ and VOC tagging

The tagging system is implemented as software which takes as input an arbitrary list of chemical species to be tagged (typically precursor emissions), and for each of these species, and arbitrary list of tags to be applied. The full suite of tagging tools, input files, and machine-readable tagged mechanism files are included in the online supplement to this manuscript. Due to the different requirements of NO$_x$ and VOC tagging, the user must explicitly choose whether NO$_x$ or VOC tagging is to be performed. The NO$_x$ tagging approach is described in Section 3.1, and the VOC tagging approach is described in Section 3.2. The resulting complete lists of both NO$_x$-tagged reactions and VOC-tagged reactions are included in both machine- and human-readable form in the supplementary material to this manuscript.

The tagging system rewrites the model chemical mechanism and CAM4 source files to include a new set of tracers and reactions corresponding to these user-specified tagged species and their associated chemical reactions. For example, if the user specifies that the tags "anthropogenic" and "biogenic" are to be applied to the species NO and NO$_2$, the chemical mechanism file will be modified to include all necessary species and reactions such that the model will be able to simulate ozone due to NO$_x$ emitted by anthropogenic and biogenic sources. The user must supply appropriate emission files containing the names of each of the tagged species in order for the additional tagged reactions and tracers to have any effect. The size of the modified chemical mechanism scales linearly with the number of tag identities requested by the user. The tagging system also modifies all model source files which contain code in which the tagged species are modified by other modelled processes such as deposition (dry and wet) and input or removal due to boundary conditions. The source code modification, including a full list of the source files which are modified, is described in more detail in Section 3.3.

Due to the potentially large number of additional reactions and species introduced into the chemical mechanism, it was necessary to modify the chemical mechanism preprocessor shipped with CESM1.2.2 to raise some hard-coded limits and ensure that the addition of the tagged reactions containing untagged species from the base mechanism does not alter the treatment of the untagged species in the chemical solver. The modified source code of the chemical preprocessor is included in the online supplement to this manuscript.

### 3.1 NO$_x$-tagged mechanism

The base chemical mechanism used here is taken from Emmons et al. (2012). The same base mechanism is used for both NO$_x$ and VOC tagging. Following Emmons et al. (2012), the reactions of tagged species are implemented as additional reactions in the model chemical mechanism file involving both tagged and untagged reactants. Untagged reactants appear in stoichio-



metrically identical amounts in the reactants and products of each tagged reaction, so that tagged reactions do not alter the concentrations of untagged species.

### 3.1.1 Separation of $NO_y$ and $O_x$ tagged species

In order to allow an arbitrary number of tags in a single model run, and to avoid the tag overwriting problem described in

Section 2, the chemical families $NO_y$ (which includes $NO_x$ and all $NO_x$ reservoir species) and $O_x$ are tagged separately. The following species from the base chemical mechanism belong to the $NO_y$ family: NO; $NO_2$; $NO_3$; $N_2O_5$; $HNO_3$; $HO_2NO_2$; $ISOPNO_3$; ONIT; ONITR; PAN; MPAN. The following species from the base chemical mechanism belong to the $O_x$ family: $O_3$; $O(^1D)$; O; $NO_2$; $NO_3$; $N_2O_5$; $HNO_3$; $HO_2NO_2$; $ISOPNO_3$; ONIT; ONITR; PAN; MPAN. When performing VOC tagging, $HO_2$ is added to the $O_x$ family (see Section 3.2.1 for more details).

Following Butler et al. (2011) we regard the reaction of NO with any peroxy radical ($HO_2$ and all organic peroxy radicals) and subsequent production of $NO_2$ as the process which effectively generates tropospheric ozone.

$$NO + HO_2 \quad \longrightarrow \quad NO_2 + OH \quad (R1)$$

Reaction R1 from the base chemical mechanism is represented in our tagging system as follows:

$$NO\_TAG + HO_2 \quad \longrightarrow \quad NO_2\_TAG + NO_2\_X\_TAG + HO_2 \quad (R2)$$

Since $NO_2$ is in both the $NO_y$ and $O_x$ chemical families, two different tagged versions of $NO_2$ are produced in Reaction R2, which represent the distinct roles of $NO_2$ in each of these chemical families: $NO_2\_TAG$ is $NO_y$-tagged $NO_2$; while $NO_2\_X\_TAG$ is $O_x$-tagged $NO_2$. The suffix "_TAG" is a placeholder which can be replaced by the tagging system with an arbitrary number of user-chosen tag identities, each of which is represented by a unique reaction added to the tagged chemical mechanism. The suffix "_X_TAG" represents members of the $O_x$ chemical family produced from emitted $NO_x$ species tagged

with the identity "TAG". Additional production pathways of $O_x$ species are discussed below.

In the base chemical mechanism, ozone is produced from $NO_2$ via photolysis:

$$NO_2 \quad \xrightarrow{h\nu} \quad NO + O \quad (R3)$$
$$O + O_2 \quad \longrightarrow \quad O_3 \quad (R4)$$

The NO produced from Reaction R3 is then available for additional reaction with a peroxy radical, while the atomic O goes on to produce $O_3$. In the tagged chemical mechanism, the fate of $O_x$-tagged $NO_2$ is different from that of $NO_y$-tagged $NO_2$:

$$NO_2\_TAG \quad \xrightarrow{h\nu} \quad NO\_TAG \quad (R5)$$
$$NO_2\_X\_TAG \quad \xrightarrow{h\nu} \quad O\_X\_TAG \quad (R6)$$
$$O\_X\_TAG + O_2 \quad \longrightarrow \quad O_3\_X\_TAG + O_2 \quad (R7)$$

In the tagged versions of Reactions R3 - R4, tagged ozone is produced from tagged $O_x$ in Reaction R6, while the tagged $O_x$ precursor NO remains available for further subsequent conversion of NO to $NO_2$ after its regeneration in Reaction R5.

The tag overwriting problem (Section 2) emerges from the reaction between ozone and NO:





$$NO + O_3 \quad \longrightarrow \quad NO_2 + O_2 \quad (R8)$$

Because Emmons et al. (2012) did not clearly distinguish the $NO_y$ and $O_x$ chemical families in their tagging system, their tagged $NO_2$ effectively inherited its tag from NO, leading to the replacement of tagged ozone identities by the $O_3$-$NO_x$ null chemical cycle. This has the effect that tag identities from nearby sources of $NO_x$ are over-represented in the tagged $O_3$ in the study of Emmons et al. (2012).

In our tagging system, we avoid this problem by handling Reaction R8 as follows:

$$NO\_TAG + O_3 \quad \longrightarrow \quad NO_2\_TAG + O_3 \quad (R9)$$
$$NO + O_3\_X\_TAG \quad \longrightarrow \quad NO_2\_X\_TAG + NO \quad (R10)$$

The tagged identity of the emitted $NO_x$ precursor is preserved in Reaction R9, while the tagged identity of $O_x$ is preserved in Reaction R10.

### 3.1.2 Preparation of the chemical mechanism for $NO_x$ tagging

Before the tagging system can automatically generate a tagged chemical mechanism file including the user-specified tag identities, a set of placeholder reactions must be added by hand to the base chemical mechanism. In future versions of our tagging system, it may be possible to identify these reactions automatically. These reactions can be classified into a number of different categories based on their chemical characteristics:

1. Reactions of emitted $NO_x$ and corresponding $NO_y$ reservoir species. This category includes all reactions between NO and peroxy radicals which generate $O_x$-tagged $NO_2$ ($NO_2\_X\_TAG$).

2. Reactions of $O_x$ species, including transformations between $O_x$ family members, and sinks of $O_x$. This category changes slightly depending on whether $NO_x$ or VOC are being tagged; for $NO_x$ tagging, reactions of OH radicals with atomic O and molecular $O_3$ are sinks of $O_x$, while for VOC tagging these reactions preserve $O_x$ (see Section 3.2.1 for more details).

3. Reactions which endogenously generate $NO_y$ or $O_x$ species. Stratospheric $O_3$ is produced in this category of reactions, through the photolysis of $O_2$ and $N_2O$ which ultimately produce the specially tagged species $O_3\_X\_STR$. A small amount of atomic O is produced from the self-reaction of OH radicals, producing the specially tagged species "$O\_X\_XTR$" ("extra" sources). This category also changes slightly depending on whether $NO_x$ or VOC are being tagged. When $NO_x$ are being tagged, photolysis of $N_2O$ in the stratosphere produces $NO\_STR$, and the reactions of $HO_2$ with certain organic peroxy radicals produce $O_3\_X\_XTR$.

Similarly to Emmons et al. (2012), the species $N_2O_5$, which is formed by reaction between $NO_2$ and $NO_3$, is duplicated to account for the possibility that its tag is inherited from either $NO_2$ or $NO_3$. The species $NO_3$ is also subject to a tag inheritance problem when being tagged as a member of the $O_x$ chemical family in the following reaction:

$$NO_2 + O_3 \quad \longrightarrow \quad NO_3 + O_2 \quad (R11)$$



In this case, the $O_x$ tag identity could be inherited from either $NO_2$ or $O_3$. Following Emmons et al. (2012) we simply let the $O_x$ tag be inherited from $NO_2$ in this case.

A full list of $NO_x$-tagged reactions is given in the supplementary material to this manuscript, including reactions producing species specially tagged as "STR" and "XTR".

## 3.2 VOC-tagged mechanism

Butler et al. (2011) introduced a methodology to recursively follow the chemistry of VOC species, starting from the emitted VOC, following all intermediate species, and ending when only unreactive products remain. For each intermediate species, additional reactions and tracers are added to the chemical mechanism which are tagged with the same identity as the originally emitted species. The added species include tagged organic peroxy radicals (generically represented here as $RO_2$, but which are explicitly tagged by our tagging system). These $RO_2$ produce $O_x$ by converting NO to $NO_2$:

$$RO_2\_TAG + NO \longrightarrow NO + NO_2\_X\_TAG + \text{tagged products} \qquad (R12)$$

The "tagged products" of Reaction R12 include tagged versions of all of the intermediate VOC associated with the corresponding reaction from the base chemical mechanism. Many such reactions also include $HO_2$ as a product, which may go on to produce $O_x$ by converting NO to $NO_2$ (Reaction R1). In order to attribute this $O_x$ production to the appropriate tag identity, $HO_2$ is included in the $O_x$ family when doing VOC tagging, and the $HO_2$ produced in tagged organic reactions is given the identity of the organic reactant responsible for its production. The $HO_2\_X\_TAG$ thus-produced gives its $O_x$ tag to $NO_2$ when reacting with NO:

$$HO_2\_X\_TAG + NO \longrightarrow NO_2\_X\_TAG + NO \qquad (R13)$$

The tagging software automatically identifies reactions involving the user-specified "primary" (or emitted) VOC species in the base chemical mechanism, and automatically generates tagged reactions of these species and their intermediates, including $NO_2\_X\_TAG$ and $HO_2\_X\_TAG$ in the products where appropriate in order to attribute production of $O_x$ to these emitted VOC species.

### 3.2.1 Preparation of the chemical mechanism for VOC tagging

In the case of VOC tagging, a number of reactions must be identified and categorised by hand, similarly to the case of $NO_x$ tagging described in Section 3.1.2.

1. Reactions involving $HO_2$. These include Reaction R13, reactions of the $HO_2$ reservoir species $HO_2NO_2$, and sinks of $HO_2$ which do not pass the tag identity onto their products (typically reactions of $HO_2$ with $RO_2$ species).

2. Reactions of $O_x$ species, including transformations between $O_x$ family members and sinks of $O_x$. This category has substantial overlap with reactions involved in $NO_x$ tagging, but with one small difference: since $HO_2$ is considered a member of the $O_x$ chemical family when tagging VOC, the production of $HO_2$ from reactions of OH radicals with




atomic O and molecular $O_3$ is not treated as a sink for $O_x$ as it is for $NO_x$ tagging (Section 3.1.2). Instead, the tagged identity is preserved as HO2_X_TAG.

3. Reactions which endogenously generate $NO_y$ or $O_x$ species. This category also has substantial overlap with $NO_x$ tagging, including the production of stratospheric $O_3$ from photolysis $O_2$. An additional reaction which is considered during VOC

tagging is the production of the specially-tagged species HO2_X_XTR from the reaction between OH and $H_2O_2$.

Following Coates and Butler (2015), the chemistry of the organic peroxy radicals in the base chemical mechanism is modified here to use the permutation approach employed by the MCM, in which the cross reactions of individual $RO_2$ are represented as unimolecular decay reactions with rates proportional to the total concentration of all $RO_2$ species. Further details are given in Coates and Butler (2015).

A full list of VOC-tagged reactions is given in the supplementary material to this manuscript, including reactions producing species specially tagged as "STR" and "XTR".

### 3.3 Automatic source code rewriting

Several of the CAM source code files must be modified in order to correctly handle the processes involving the tagged tracers. Source files are first modified by hand in such a way that they can be automatically rewritten by the tagging software to

accommodate the tagged tracers, and will also compile and run correctly when the CAM is run without tagging enabled. This is accomplished by enclosing sections of relevant code between FORTRAN comments. For example, model variables which index the concentration array for tagged species are declared as follows:

```
! START TAGGING CODE
integer :: no_tag_ndx, no2_tag_ndx, no2_x_tag_ndx
! END TAGGING CODE
```

The tagging logic itself is similarly enclosed between comments. The tagging software scans each source file for these comment lines, and expands the code where appropriate, adding code for each tagged tracer which has been added to the chemical mechanism.

A full list of VOC-tagged reactions is given in the supplementary material to this manuscript, including reactions producing

The modified files are listed here, along with short summaries of the changes made in each case. The hand-modified source

files themselves, along with the tagging software and all other necessary input files are available in the supplementary material to this manuscript.

– cam_history.F90 Code is modified to account for the larger number of tracers which could potentially be written to history files.

– mo_aerosols.F90 Code for gas/aerosol partitioning of tag identities between ammonium nitrate and nitric acid is added.

– mo_airplane.F90 Code is added to tag emissions from aircraft with the hard-coded identity "AIR".





- mo_drydep.F90 Dry deposition fluxes are calculated for tagged species using deposition velocities of the corresponding untagged species.

- mo_flbc.F90 Species added at the lower model boundary are appropriately tagged if tags are defined for these species.

- mo_fstrat.F90 Tagged tracers are adjusted at the upper model boundary based on the adjustments made to the corresponding non-tagged species. Any $O_x$ or $NO_y$ added to the model is tagged as being of stratospheric origin. Other species are added or removed in proportion to their share of the corresponding untagged species.

- mo_gas_phase_chemdr.F90 Indices into the model concentration array for tagged species are determined during initialisation.

- mo_imp_sol.F90 Relative error parameters for tagged species in the implicit solver are set to the same values as for the corresponding untagged species.

- mo_lightning.F90 Code is added to tag NO production from lightning with the hard-coded identity "LGT".

- mo_neu_wetdep.F90 Wet deposition fluxes are calculated for tagged species using removal rates of the corresponding untagged species.

- mo_photo.F90 Photolysis rates for the tagged reactions are set equal to the corresponding untagged reactions.

- mo_setext.F90 Code is added to facilitate the tagging of lightning NO and aircraft emissions.

- mo_sethet.F90 Loss rates due to heterogeneous chemistry are calculated for tagged species using removal rates of the corresponding untagged species.

- mo_srf_emissions.F90 Emissions of isoprene and monoterpenes are tagged appropriately if tags have been specified for these species.

- mo_usrrxt.F90 Rate constants of several of the tagged reactions are set equal to the rate constants of the corresponding untagged reactions.

## 4  Experiment Design

We use CESM version 1.2.2 (Tilmes et al., 2015; Lamarque et al., 2012) with the component set "FSDCHM" at a horizontal resolution of $1.9 \times 2.5$ degrees, with 56 vertical levels. This component set includes the tropospheric chemistry version of CAM4-chem forced with specified dynamics from year 2010 of the MERRA reanalysis (Rienecker et al., 2011). $NO_x$, $O_3$, $HNO_3$, $N_2O_5$, $N_2O$, CO, and $CH_4$ are relaxed towards climatological values in the stratosphere. For this study, we replace the default chemical mechanism with the base mechanism from Emmons et al. (2012), modified as described in Section 3. Emissions of anthropogenic species are taken from the HTAP_v2.2 emission inventory (Janssens-Maenhout et al., 2015).



Biomass burning emissions are from GFEDv3 (van der Werf et al., 2010). Emissions of $NO_x$ from lightning are calculated online within the model according to Price et al. (1997). Biogenic emissions of $NO_x$ (from soils) and VOC (from vegetation) are prescribed as in Tilmes et al. (2015). Mixing ratios of $CH_4$ and $N_2O$ are fixed at the surface as in Tilmes et al. (2015).

Model runs are done using both $NO_x$ and VOC tagging, with the base chemical mechanism and model source code modified in each case as described in Section 3. We specify separate tag identities for emissions from anthropogenic (ANT), biogenic (BIO), biomass burning (BMB), and aircraft (AIR) sources. For $NO_x$ tagging runs we specify an additional tag for $NO_x$ from lightning (LGT), and for VOC tagging runs we specify an additional tag for methane (CH4). In both cases ($NO_x$ and VOC tagging) we include tags representing chemical production in the stratosphere (STR), "extra" chemical production (XTR, as described in Section 3), and a special tag representing the initial conditions (INI), allowing us to monitor the progress of the model spinup. This choice of tag identities allows us to compare our source attribution with that of Emmons et al. (2012), who used a similar set of tag identities, and on which our new tagging scheme is based.

Initial conditions for $O_x$ species were tagged with STR in the stratosphere, and INI in the troposphere. Following Emmons et al. (2012), we used a chemical tropopause definition of 150 ppb of ozone. Initial methane in the VOC-tagging run was tagged with CH4. The concentration of INI-tagged and STR-tagged species was set equal to the mixing ratio of the corresponding species in the initial conditions, and all other tagged tracers were set to zero at the beginning of the model run. The model was run with annually repeating meteorology from 2010 until the maximum contribution of surface ozone attributable to the initial conditions was less than 1% of the total surface ozone, and the maximum difference between the stratospheric contribution to surface ozone in December from the stratospheric contribution to surface ozone in the previous December was also less than 1%. For VOC tagging we imposed the additional constraint that the difference between the contribution of methane to surface ozone in December and the contribution in the previous December was less than 1%. This was achieved after 2 years of simulation for $NO_x$-tagged runs, and 3 years of simulation for VOC-tagged runs. For the final year of simulation win each case, we verified that the method was working as expected by comparing the sum of the tagged ozone tracers with the actual ozone simulated by the model. At the lowest model level, the maximum monthly average difference was of the order of $1 \times 10^{-5}$ ppb, while in the free troposphere the maximum monthly average difference was of the order of 1 ppb. The final year of simulation for both $NO_x$- and VOC-tagged runs is presented and discussed in Section 5.

## 5 Results

January average surface ozone mixing ratio, along with the mixing ratios of major contributing sources are shown from the $NO_x$-tagging run in Figure 1 and from the VOC-tagging run in Figure 2. Similarly, July average surface ozone mixing ratio is shown for $NO_x$- and VOC-tagging in Figures 3 and 4.

In the northern mid-latitudes, the land-sea gradient of modelled surface ozone reverses sign between January and July. Over the mid-latitude continental regions, modelled surface ozone has its maximum in summer, and its minimum in winter. Over the remote ocean regions, the opposite is the case; modelled surface ozone concentrations are higher in winter than they are in summer. Low modelled surface ozone mixing ratios over the northern mid-latitudes in winter are consistent with high



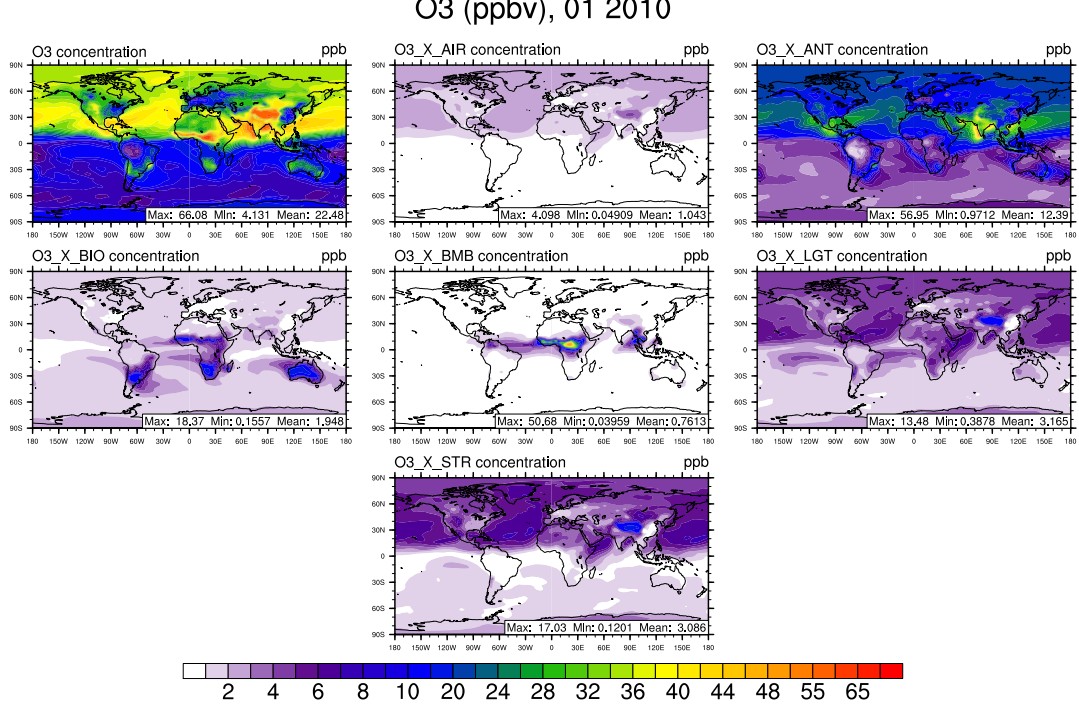

**Figure 1.** Surface ozone in January from the NO$_x$-tagging run. Total surface ozone is shown in the top-left panel. Other panels show the contribution to surface ozone due to NO$_x$ precursors emitted by aircraft (AIR), anthropogenic sources (ANT), biogenic sources (BIO), biomass burning (BMB), lightning (LGT), and transport from the stratosphere (STR).

local emissions of NO$_x$, and ozone removal by Reaction R8. High modelled surface ozone mixing ratios over the northern mid-latitudes in summer are primarily attributable to a combination of anthropogenic NO$_x$ emissions and biogenic NMVOC emissions, combined with more active photochemistry due to higher insolation. Anthropogenic NMVOC contribute relatively little to modelled high surface ozone mixing ratios in the boreal summer. This difference is consistent with the relatively high

5    reactivity of biogenic NMVOC, especially isoprene, as well as the strong seasonal cycle in biogenic NMVOC emissions in mid-latitude regions, being emitted almost exclusively during the growing season.

Low modelled surface ozone mixing ratios over the remote northern hemispheric ocean regions in summer are consistent with a stronger chemical sink due to photolysis of ozone with subsequent production of OH radicals from water vapor (Johnson et al., 1999). The strength of this sink decreases during the winter, allowing modelled ozone to build up over large regions of the

10    remote northern hemisphere. This hemispheric background ozone reaches a maximum in March/April (not shown) before the




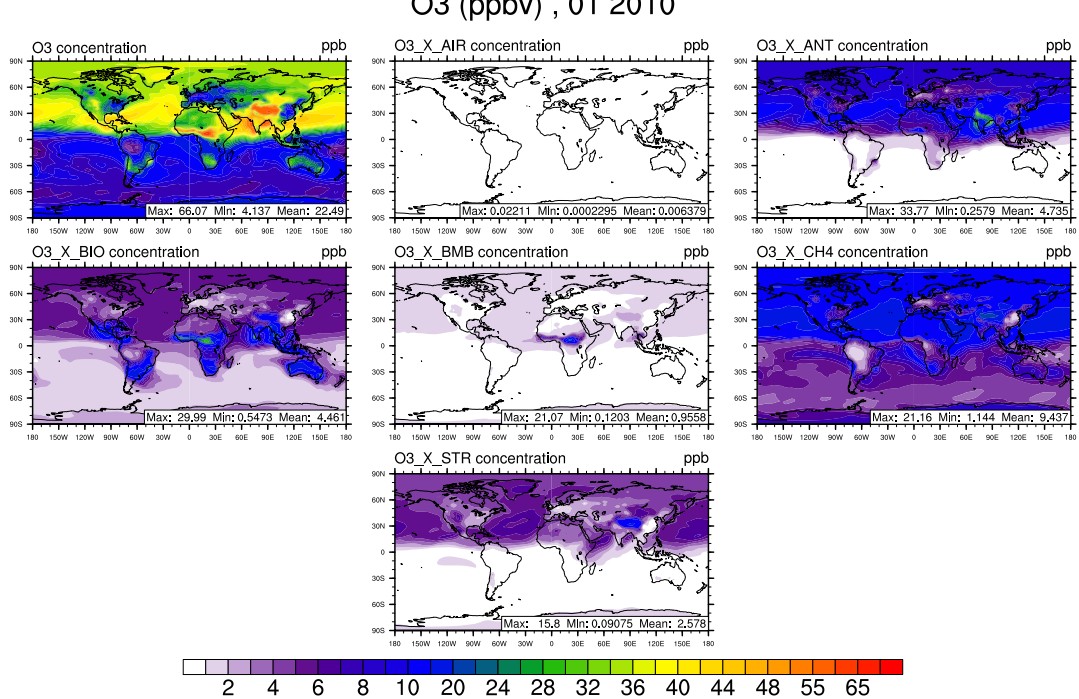

**Figure 2.** Surface ozone in January from the VOC-tagging run. Total surface ozone is shown in the top-left panel. Other panels show the contribution to surface ozone due to organic precursors emitted by aircraft (AIR), anthropogenic sources (ANT), biogenic sources (BIO), biomass burning (BMB), methane (CH4), and transport from the stratosphere (STR).

chemical sink increases again. Examination of the tagged ozone tracers shows that this winter-spring remote maritime buildup of ozone is primarily attributable to both anthropogenic $NO_x$ and NMVOC emissions. This is in contrast to the summer maximum in surface ozone modelled over continental regions, for which the primary responsible NMVOC precursor is of biogenic origin. A strong sensitivity of tropospheric ozone to anthropogenic $NO_x$ and biogenic VOC emissions has been

5 noted in previous studies (eg. Young et al., 2013; Stevenson et al., 2013), but we are not aware of any previous work in the peer reviewed literature showing that anthropogenic non-methane VOC contribute disproportionately to springtime ozone over remote regions of the northern hemisphere.

Another noteworthy feature of Figures 2 and 4 is the strong contribution of methane to the modelled mixing ratio of ozone at the surface, in both January and July. A strong sensitivity of modelled tropospheric ozone to the mixing ratio of methane has

10 been noted in previous work (eg. Fiore et al., 2008; Young et al., 2013). Here, we show that the contribution of surface ozone



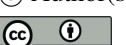

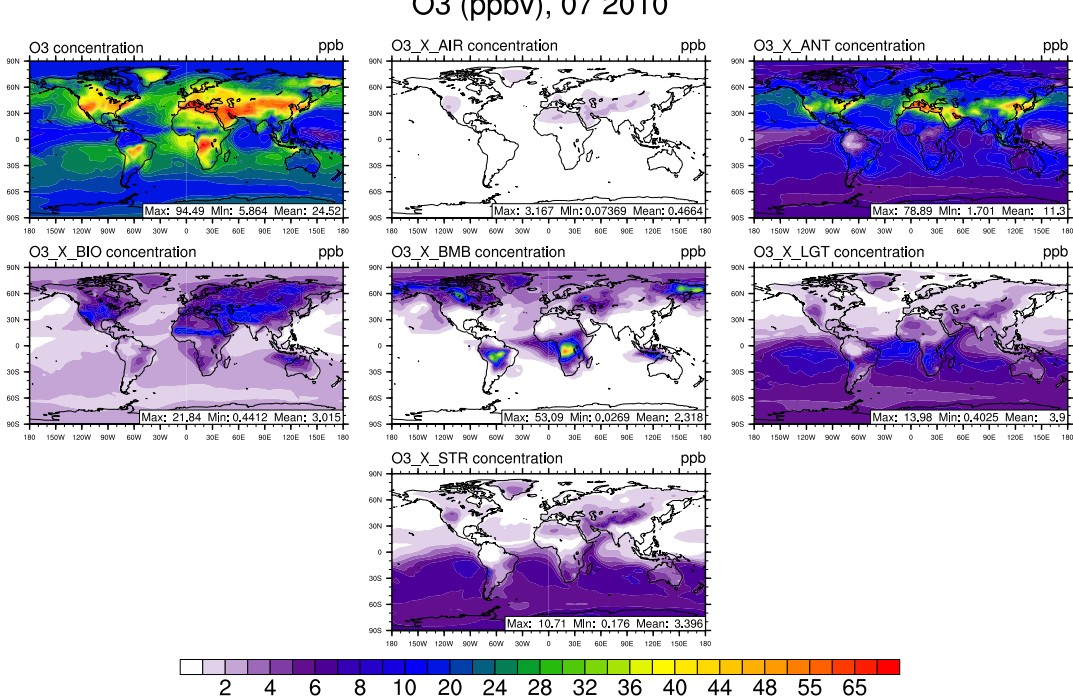

**Figure 3.** Surface ozone in July from the $NO_x$-tagging run. Total surface ozone is shown in the top-left panel. Other panels show the contribution to surface ozone due to $NO_x$ precursors emitted by aircraft (AIR), anthropogenic sources (ANT), biogenic sources (BIO), biomass burning (BMB), lightning (LGT), and transport from the stratosphere (STR).

attributable to methane as an organic precursor remains remarkably constant at about 15 ppb over large regions of the northern hemisphere year-round (at least in our model).

The influence of the stratosphere on the modelled ozone mixing ratio at the surface is stronger in winter than in summer. The stratospheric influence on northern hemisphere surface ozone is smallest in July and August, and reaches a maximum in March 5 (not shown), when the contributions from the stratosphere, and the organic precursors methane and anthropogenic VOCs to the northern hemispheric background ozone are approximately equal. The late-winter early-spring maximum in the stratospheric contribution to surface ozone is consistent with both an increased lifetime of tropospheric ozone during this period, as well as the increasing flux of ozone from the stratosphere, which is consistent with the earlier work of Roelofs and Lelieveld (1997), who also used a stratospheric ozone tracer to determine the contribution of the stratosphere to surface ozone.



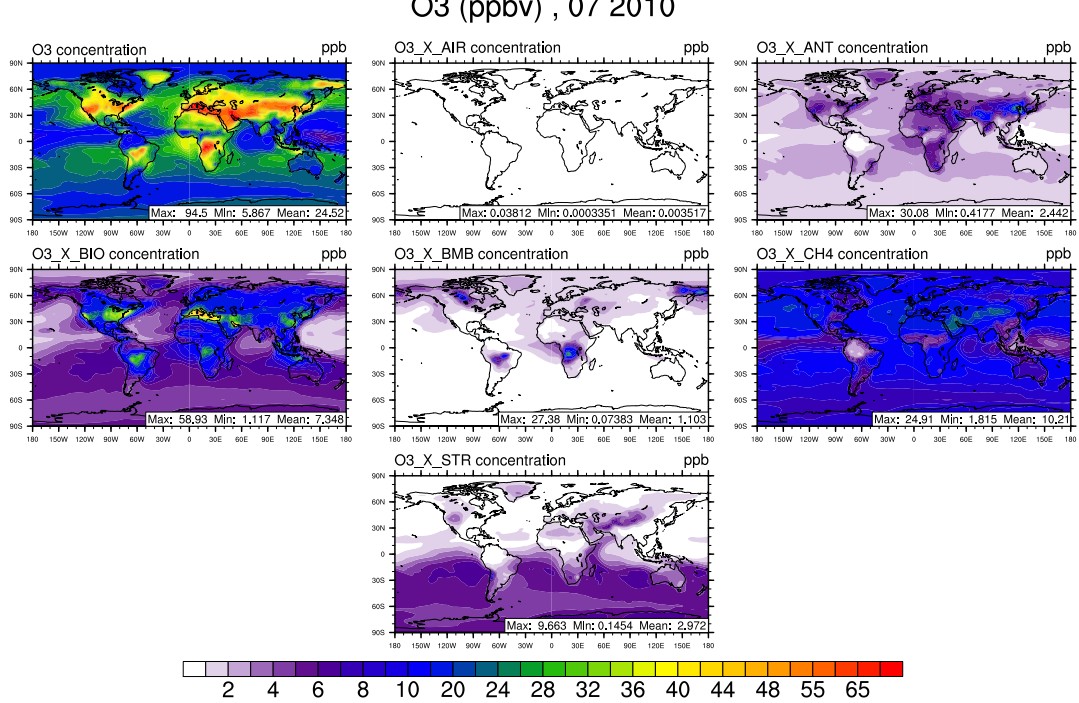

**Figure 4.** Surface ozone in July from the VOC-tagging run. Total surface ozone is shown in the top-left panel. Other panels show the contribution to surface ozone due to organic precursors emitted by aircraft (AIR), anthropogenic sources (ANT), biogenic sources (BIO), biomass burning (BMB), methane (CH4), and transport from the stratosphere (STR).

Emmons et al. (2012) determined the contribution of stratospheric ozone to the modelled mixing ratio of ozone at the surface using their tagging approach. Since they did not explicitly tag the ozone originating in the stratosphere, they calculated the stratospheric contribution to tropospheric ozone as the residual after subtracting all of the ozone which had been produced from tagged tropospheric sources. They found that their residual stratospheric contribution to surface ozone was less than half of the contribution determined using a stratospheric tracer. Emmons et al. (2012) pointed out that such a stratospheric ozone tracer is likely to give an upper bound on the stratospheric contribution to surface ozone due to the fact that the tagged stratospheric ozone is set equal to the total ozone mixing ratio in the stratosphere, which effectively overwrites any tropospheric ozone which may have been imported into the stratosphere. We also regard the residual estimate of Emmons et al. (2012) as a lower bound on the contribution of stratospheric ozone to surface ozone, due to the "overwriting" problem mentioned above, in which their ozone tag identities are overwritten with the identity of nearby sources.





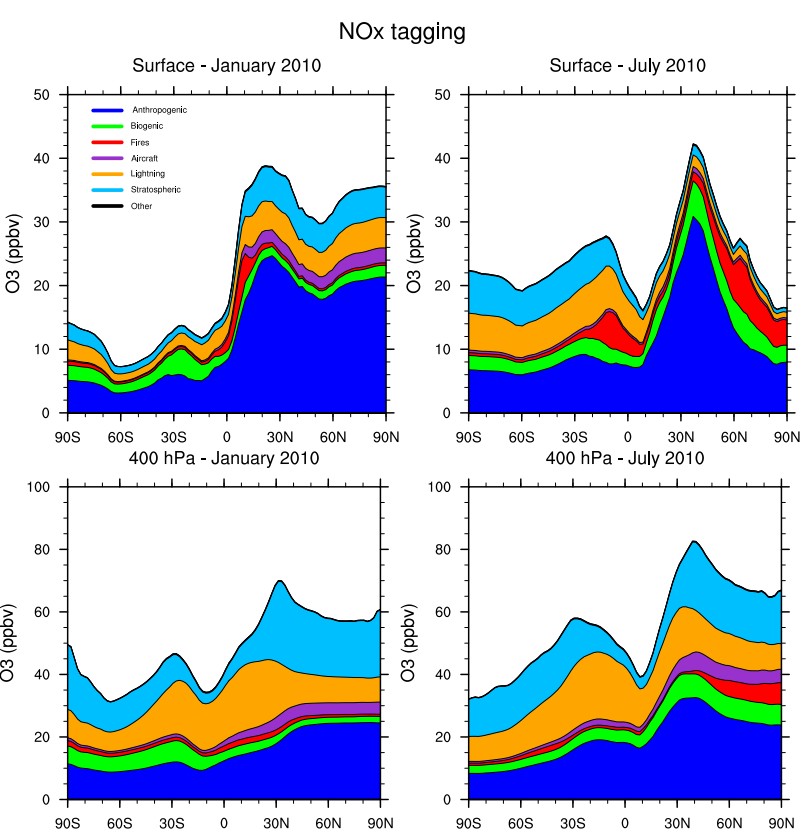

**Figure 5.** Zonal average of tagged ozone source contributions at the surface (top panels) and at 400 hPa (bottom panels) for January (left panels) and July (right panels) from the NO$_x$-tagging run





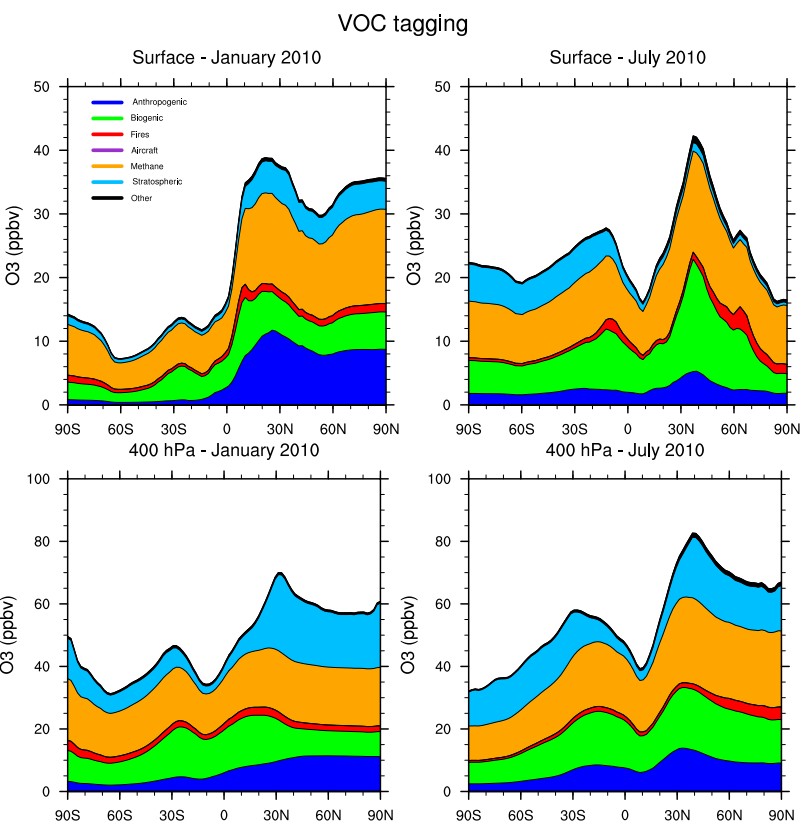

**Figure 6.** Zonal average of tagged ozone source contributions at the surface (top panels) and at 400 hPa (bottom panels) for January (left panels) and July (right panels) from the VOC-tagging run



Figure 5 shows the contribution of each of our tag identities to the zonally averaged ozone at the surface and at 400 hPa from our NO$_x$ tagging run. This figure is designed to be directly comparable with Figure 6 of Emmons et al. (2012). Our simulated zonal average total ozone mixing ratio is broadly similar with that of Emmons et al. (2012) in both January and July, but there are some noteworthy differences in contributions of the tagged tracers; the stratospheric contribution to surface ozone shows

particularly large differences. We model a zonally averaged stratospheric contribution to surface ozone of approximately 8 ppb in each winter hemisphere (NH in January and SH in July). These results are similar to those of Emmons et al. (2012) in the southern hemisphere, but approximately double those in the northern hemisphere winter, where Emmons et al. (2012) attribute only about 4 ppb of surface ozone to stratospheric origin. The lower stratospheric contribution to northern hemisphere surface ozone from Emmons et al. (2012) is consistent with their bias towards nearby sources due to the tag overwriting problem,

as noted above. Similarly, Emmons et al. (2012) estimate a higher (by approximately 5 ppb) contribution of anthropogenic emissions to zonal average surface ozone than we see in our Figure 5, and show effectively no stratospheric contribution in July, while our run shows a small contribution of about 3 ppb stratospheric ozone to surface ozone in July. These results illustrate the importance of explicitly separating tagged species which are members of both the NO$_y$ and O$_x$ families to preserve the tagged identities of ozone transported over long distances.

The contribution of the tagged VOC precursors to zonal average surface ozone is shown in Figure 6. The widespread, year-round contribution of methane to ozone production is clearly visible, as is the increased importance of anthropogenic non-methane VOC as an ozone precursor during winter, noted earlier. Figure 6 also includes a contribution from the stratospherically tagged ozone in our VOC tagging run. It is immediately apparent from comparison with Figure 5 that the stratospheric contribution to tropospheric ozone is lower in the VOC tagging run than in the NO$_x$ tagging run. Since the direct

production of stratosphere-tagged ozone is identical in both runs, this difference must be due to ozone production involving stratosphere-tagged NO, produced from photolysis of N$_2$O as described in Section 3.1.2. By comparison with Figure 5, we see that this NO$_x$ from the stratosphere contributes approximately an additional 2 ppb to the surface ozone ultimately attributable to the stratosphere (or approximately one quarter of the total stratospheric contribution). We are not aware of any previous work quantifying the contribution of the photolysis of N$_2$O in the stratosphere to the photochemical production of ozone in

the troposphere. We note that our model does not include a comprehensive treatment of stratospheric chemistry and associated stratosphere-troposphere exchange. While our model does explicitly represent the photolysis of O$_2$ and N$_2$O in the stratosphere, the mixing ratios of O$_x$ and NO$_y$ species are also relaxed towards climatological values in the stratosphere. Future work examining the contribution of stratospheric NO$_x$ to tropospheric ozone production should implement our tagging methodology in a fully coupled stratosphere-troposphere model.

**6 Conclusions**

We have introduced and described a technique for attribution of tropospheric ozone to emitted precursors of both NO$_x$ and VOC, as well as transport from the stratosphere. The results obtained using this technique are consistent with understanding of tropospheric ozone chemistry based on previous work. Our work shares features with many earlier methodologies for





attribution of tropospheric ozone, but combines these features in unique ways which allow a unique and deeper understanding of the processes influencing tropospheic ozone in our model, and avoid many of the problems associated with previous work such as over-attribution of ozone to locally emitted precursors, and the unphysical transfer of tag identities between $NO_x$ and VOC species.

By performing simultaneous but separate attribution of ozone to both its $NO_x$ and VOC precursors, we have quantified, for example, the changing contributions of anthropogenic and biogenic sources to modelled seasonal cycles of surface ozone over the populated and remote regions of the northern hemisphere. In particular, we have identified the combination of anthropogenic $NO_x$ and anthropogenic VOC as a significant contributor to the widespread buildup of ozone over the northern hemisphere during winter-spring in our model, in contrast with a relatively insignificant role for anthropogenic VOC in sum-

mer ozone production, for which biogenic VOC play a more important role. Further experiments using this tagging technique should examine the winter-spring contribution of anthropogenic VOC in more detail. Such experiments could instead tag anthropogenic VOC emissions according to their source sector, geographical region, time of emission, or even according to the particular kinds of VOC molecules emitted, in order to understand more about the ultimate sources of this springtime ozone in different receptor regions.

Given the problems of the current generation of global chemistry-climate models in simulating amounts, trends, and seasonal cycles of tropospheric ozone, the deeper understanding provided by our tagging methodology may yield information about deficiencies in these models and point the way towards improvements. If implemented in additional chemistry-climate models, our methodology could be a useful tool in understanding the differing responses of different models to changes in precursor emissions Given the large number of alternative methodologies for attribution of tropospheric ozone, including the several

different ways of implementing tagging which have been reviewed here, we also believe that the community would benefit from a systematic intercomparison of the different techniques for constructing source-receptor relationships of tropospheric ozone.

*Code availability.* The full suite of tagging tools, input files, and machine-readable tagged mechanism files are included in the supplementary material to this manuscript.

*Author contributions.* TB conceived and designed the study. TB implemented the automatic mechanism rewriting and code generation tools. ZS and AL adapted the CESM source code. AL performed the model runs and subsequent analysis. AL and JC both contributed tools for analysing the model runs. TB wrote the paper.

*Competing interests.* The authors declare no competing interests.





*Acknowledgements.* This work was hosted by IASS Potsdam, with financial support provided by the Federal Ministry of Education and Research of Germany (BMBF) and the Ministry for Science, Research and Culture of the State of Brandenburg (MWFK).




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
