# Peer review of "TOAST 1.0: Tropospheric Ozone Attribution of Sources with Tagging for CESM 1.2.2"

_Geoscientific Model Development, 2018_

## Referee Comment (RC1) · Anonymous Referee #1 · 5 Apr 2018

The tagging introduction was particularly well-done and, by itself, is a nice contribution. The automated tagging system is also significant, but it is not totally clear what fraction of the work is automated versus manual in the text. The CH4 contribution and stratospheric N2O results are particularly valuable.

Overall, I had very few questions or comments, which is rare. My one concern is the level of detail provided about the manual updates for a new set of tags.

Line-by-line:

pg3,18-19 : The Sillman paper has an appendix where they derive the ratio of 0.5 (not 0.35). The 0.35 was an approximation when using a chemical mechanism that did not include some loss pathways (e.g., ROOH). It would be nice to be more clear about this.

[Figure]

pg4,5 : This reviewers understanding is that NO and RO2 are not tagged, but NO and RO2 reactions are apportioned proportional to NOy and NMHC.

pg4,17-21; pg5,1-4 : Interesting thing to note. Does your NO2_X_TAG ever react with RO2s? If so, could it make PAN and thus suffer a similar problem?

pg6,7 : and should be an?

pg9,1-2 : Given that NO3 has two odd-oxygens, why not attribute 1/2 to each?

pg12,21 :win?
* * *

---

## Referee Comment (RC2) · Anonymous Referee #2 · 20 Apr 2018

Manuscript Summary

The manuscript presents a well written summary of the implementation of a tracer tagging system within the Community Earth System Model that is able to identify and track the sources of tropospheric ozone. Separate tagging schemes are available for NOx and VOC precursor emissions which avoids some of the pit falls from previous schemes. An example of the using the tagging scheme is presented and highlights the ability of the scheme to identify the contribution of different of sources to ozone formation. I found the paper well written and suitably detailed and think would make a valuable contribution to future source-receptor studies. I would recommend publication once the following comments have been addressed.

General Comments

1. On a number of occasions, in the introduction and conclusions, the manuscript mentions that tagging methods are complimentary to perturbation (sensitivity) methodologies for analysing source-receptor relationships. Perhaps the author would like to comment further in the conclusion sections on how might this be achieved and what particular aspects of the two different methods are complimentary or comparable.

2. As expected the manuscript focussed on the reactions involved in the production of ozone. However, there is not much mention made about termination/loss reactions and how these interact with the tagging scheme. It is mentioned in Section 3 (e.g. Section 3.2.1 P9) but it would be good to mention this a bit more and perhaps provide an example reaction of how tagging is treated in these reactions (or refer to the supplementary).

3. Is there a clearer way of labelling the tagged tracers to make then more identifiable with their source? For example the Ox tagged tracers have the suffix '_X_TAG' whereas the NOy tagged tracers are labelled as '_TAG'. Could the NOy tagged tracers not be labelled as '_Y_TAG' or '_N_TAG' to clearly identify their source?

4. Throughout section 5 there are numerous times when winter is mentioned in isolation (e.g. P12 Line 31). Please could the author check in the results section that reference is made to northern hemisphere winter or just the individual month to avoid confusion.

5. Throughout section 5 there are numerous references to results in March/April or Spring. However, no such results are presented in the manuscript. I found it very frustrating for the manuscript to be talking about results which I could not see. Therefore I would like to see the results presented for this season in order to be able to confirm any assertions made in the manuscript.

6. Key on Figure 5 and 6 are quite small. These figures could be enlarged for the final version or the keys made larger to make sure that they are clear and legible.

Minor Comments

Section 1. P1, Line 16 – 'as well as an contributor' should be changed to 'as well as a contributor'.

Section 2. P5. Lines 22-25 and Lines 1 to 4 on P6. – Perhaps this whole paragraph would be better placed as the introduction to the methods section since it is highlighting the improvements from the scheme developed here.

Section 3. P6 Line 7 – 'and arbitrary list of tags to be applied' replaced with 'an arbitrary list of tags to be applied'.

Section 3. P6 Line 15 – Is it possible here or in the supplementary to supply a list of possible tags that could be applied and also list what emission files are required to be provided for each tag. Tags are mentioned in Section 4 but perhaps could be brought forward to here as well.

Section 3.1.1. P7 and 8 – I found the description of how to avoiding over-representing the influence from local NOx sources using the separate tagged tracers a little bit confusing at times. Could this mechanism possibly be represented schematically to help the user in tracking the different pathways that the tagged tracers follow?

Section 3.1.2. P8 Lines 11 to 12 – Are these manual reactions separate to the automatically determined ones and could they be separately flagged in the supplementary material?

Section 3.1.2 P9 Lines 1 to 2 – How much does letting the Ox tag be inherited from NO2 sources impact on the tagging scheme (related to point 2 above).

Section 4. P12 Line 21 – 'win' should be 'in'

Section 4. P12 Line 21 and 24– Final year of the simulation is mentioned whereas it would be nice to state actual year of the simulation in which results can be obtained (i.e. 4th year).

Section 4. P12 Lines 23 to 24 – Is there a reason that using the tagged chemical mechanism generates different results to the original mechanism, particularly in the free troposphere?

Section 5. P12 Line 30 – I don't think that the gradient really reverses that much for ozone from anthropogenic sources, changes are more subtle.

Section 5. P14. Lines 2 to 4 – Is this sentence talking about O3 only from the VOC tagging? I think it this needs to be clarified in the sentence.

Section 5. P15 Line 1 – I found the colours on these Figures quite hard to determine actual concentrations from. The blues seem to cover the range of between 8 and 20 ppb making it hard to identify precisely the contribution from methane. Would using a different colour scale (or different increments) provide better results?

Section 5. P 15 Line 3 – Is the influence of the stratosphere stronger in winter, looks like just a shift in hemispheres.

Section 5. P 15 Line 4 – August is mentioned but no results shown to verify (See point 5 above).

Section 5. P 16 Lines 6 to 9 – In one sentence mention is made of an upper bound on the stratospheric contribution whilst later on it is referred to as a lower bound. Please could the author clarify if this is correct.

Section 5. P 19 Line 17 to 18 – Could you provide numbers to verify that the stratospheric contribution is lower in the VOC tagging than for NOx tagging.

Section 5. P19 Lines 25 to 29 – Mention is made here of the limitations in the stratospheric chemistry within the model. I think it would be useful to briefly mention if anticipated future improvements to stratospheric chemistry are likely to increase or reduce the stratospheric contribution to tropospheric ozone.

[Figure]

2018.
Interactive
comment

---

## Short Comment (SC1) · 1 May 2018

This is a great paper and I am happy to see more work on tagging techniques at the global scale which I believe could really help us understand tropospheric ozone budgets and model diversity.

The initial work presented with this technique is very interesting. I wonder if the authors would consider adding a table with the quantitative contributions of the different tagged tracers to the ozone burden? These would be useful numbers to have in the literature.

Colette Heald, MIT

2018.

---

## Author Comment (AC1) · 5 Jun 2018

**Authors' comment in response to all comments made in the open discussion phase**

We thank the two Anonymous Referees and Colette Heald for their thoughtful comments on our submission. We have revised our manuscript taking all of these comments into account. We believe that this process has significantly improved the manuscript. Here we reproduce the comments in *italic font*, and in each case provide our responses and a summary of the resulting changes to the manuscript (if any) in normal font. We also append a copy of the revised version of the manuscript with the changes highlighted to the end of this author comment.

**Response to Anonymous Referee #1**

*The tagging introduction was particularly well-done and, by itself, is a nice contribution. The automated tagging system is also significant, but it is not totally clear what fraction of the work is automated versus manual in the text. The CH4 contribution and stratospheric N2O results are particularly valuable. Overall, I had very few questions or comments, which is rare. My one concern is the level of detail provided about the manual updates for a new set of tags.*

We thank the referee for their positive review. To address their concern about the process of adding new tags, we have restructured Section 3, adding more detail about the extent to which this process is automated, clarifying that the implementation of a new set of tags is an automated process once some initial modifications have been made to the chemical mechanism and model source files. The necessary modifications were already described in the original version of the manuscript. Please note that the restructuring of Section 3 is also in response to a comment from Referee #2 regarding the possible choices of tags.

*pg3,18-19 : The Sillman paper has an appendix where they derive the ratio of 0.5 (not 0.35). The 0.35 was an approximation when using a chemical mechanism that did not include some loss pathways (e.g., ROOH). It would be nice to be more clear about this.*

We have expanded the discussion of Sillman (1995) to include the distinction between transition thresholds based on total peroxides and hydrogen peroxide.

*pg4,5 : This reviewers understanding is that NO and RO2 are not tagged, but NO and RO2 reactions are apportioned proportional to NOy and NMHC.*

The reviewer is correct. We have updated our discussion of Grewe et al. (2017) to state that the attribution of tags is done proportionally.

*pg4,17-21; pg5,1-4 : Interesting thing to note. Does your NO2_X_TAG ever react with RO2s? If so, could it make PAN and thus suffer a similar problem?*

Yes; and no. Our NO2_X_TAG can react with RO2s to make PAN_X_TAG, but upon decomposition, this PAN_X_TAG simply regenerates NO2_X_TAG (and no organic products), so does not suffer from this problem. The relevant reactions are already shown in the supplementary material. We have added a short sentence to the end of Section 3.1.1 referring the reader to the supplement for further details of the chemistry of tagged species.

*pg6,7 : and should be an?*

Correct. We have fixed this typo.

*pg9,1-2 : Given that NO3 has two odd-oxygens, why not attribute 1/2 to each?*

This suggestion by the referee is probably more correct than what we have done. We will consider implementing this change in a future version of the tagging system.

*pg12,21 :win?*

This should read "in", and has been fixed.

**Response to Anonymous Referee #2**

*The manuscript presents a well written summary of the implementation of a tracer tagging system within the Community Earth System Model that is able to identify and track the sources of tropospheric ozone. Separate tagging schemes are available for NOx and VOC precursor emissions which avoids some of the pit falls from previous schemes. An example of the using the tagging scheme is presented and highlights the ability of the scheme to identify the contribution of different of sources to ozone formation. I found the paper well written and suitably detailed and think would make a valuable contribution to future source-receptor studies. I would recommend publication once the following comments have been addressed.*

We thank the referee for their positive assessment of our manuscript. We address the individual comments below.

*1. On a number of occasions, in the introduction and conclusions, the manuscript mentions that tagging methods are complimentary to perturbation (sensitivity) methodologies for analysing source-receptor relationships. Perhaps the author would like to comment further in the conclusion sections on how might this be achieved and what particular aspects of the two different methods are complimentary or comparable.*

We believe that the introduction to the paper is the appropriate place to go into further detail on the complementary nature of tagging and perturbation, so we have expanded the discussion on this in the third paragraph of the introduction. As requested by the referee, we have also added text to the conclusions giving an example of a possible application of combining tagging and perturbation: quantifying the change in the contribution of methane to modelled tropospheric ozone when anthropogenic NOx emissions are changed by some amount.

*2. As expected the manuscript focussed on the reactions involved in the production of ozone. However, there is not much mention made about termination/loss reactions and how these interact with the tagging scheme. It is mentioned in Section 3 (e.g. Section 3.2.1 P9) but it would be good to mention this a bit more and perhaps provide an example reaction of how tagging is treated in these reactions (or refer to the supplementary).*

The reviewer is correct that we have focused more on production pathways in our description of the tagging system. In order to remedy this, we have added an example of the $O(^1D) + H_2O$ reaction as a loss process for tagged $O_x$ to the end of Section 3.1.1. We also note that extra text has also been added here referring to the supplement as part of our response to Referee #1.

*3. Is there a clearer way of labelling the tagged tracers to make then more identifiable with their source? For example the Ox tagged tracers have the suffix '_X_TAG' whereas the NOy tagged tracers are labelled as '_TAG'. Could the NOy tagged tracers not be labelled as '_Y_TAG' or '_N_TAG' to clearly identify their source?*

Yes, this would be possible, but we do not see the need for this. In the system as it is presently implemented, the suffix "_TAG" is applied to emitted precursors and their chemical products (either VOC or NOx depending on which type of tagging is being used), and the suffix "_X_TAG" is used for the $O_x$ family. Since "TAG" is just a placeholder which is replaced by an arbitrary set of user-defined tags, the user of the system is free to choose names for their tags which make the most sense to them.

*4. Throughout section 5 there are numerous times when winter is mentioned in isolation (e.g. P12 Line 31). Please could the author check in the results section that reference is made to northern hemisphere winter or just the individual month to avoid confusion.*

We have gone through Section 5 and made sure that it is clear which hemisphere is being referred to.

*5. Throughout section 5 there are numerous references to results in March/April or Spring. However, no such results are presented in the manuscript. I found it very frustrating for the manuscript to be talking about results which I could not see. Therefore I would like to see the results presented for this season in order to be able to confirm any assertions made in the manuscript.*

We have added figures analogous to Figures 1, 2, 3, and 4 for each month of the year to the online supplement, for both $NO_x$ and VOC tagging. This results in 24 figures, each with 7 panels. This would be too many figures to include in the main manuscript. Including them in the supplement allows all interested readers to verify our assertions, and explore our results for themselves in more depth. To complement this, we have added text near the beginning of the results section pointing the reader at these additional figures, and have also referred to the supplement when we discuss months other than January or June.

*6. Key on Figure 5 and 6 are quite small. These figures could be enlarged for the final version or the keys made larger to make sure that they are clear and legible.*

We have enlarged the keys in Figures 5 and 6.

*Minor Comments*

*Section 1. P1, Line 16 – 'as well as an contributor' should be changed to 'as well as a contributor'.*

We have fixed this typo.

*Section 2. P5. Lines 22-25 and Lines 1 to 4 on P6. – Perhaps this whole paragraph would be better placed as the introduction to the methods section since it is highlighting the improvements from the scheme developed here.*

We agree with the reviewer on this point, and have made the requested change.

*Section 3. P6 Line 7 – 'and arbitrary list of tags to be applied' replaced with 'an arbitrary list of tags to be applied'.*

We have fixed this typo.

*Section 3. P6 Line 15 – Is it possible here or in the supplementary to supply a list of possible tags that could be applied and also list what emission files are required to be provided for each tag. Tags are mentioned in Section 4 but perhaps could be brought forward to here as well.*

It seems that we could have been clearer about how flexible our tagging system is. The choice of tags is completely up to the user of the system, and could be totally different depending on the application. For example, the tags could be based on source sectors (the example we gave), geographical regions, particular time periods, or really just about anything the user can dream up. In each case, however, the user must specify appropriate emission files, since it is well beyond the scope of the automatic tagging system to anticipate all possible use cases. We have restructured the Section 3 in order to make this clearer. Please note that this restructuring is also in response to a comment by Referee #1 regarding the extent to which the tagging is automated.

*Section 3.1.1. P7 and 8 – I found the description of how to avoiding over-representing the influence from local NOx sources using the separate tagged tracers a little bit confusing at times. Could this mechanism possibly be represented schematically to help the user in tracking the different pathways that the tagged tracers follow?*

We agree with the Referee that it would be nice to have a clear schematic diagram of how tag identities are transferred between tracers in our approach. Unfortunately it is difficult to come up with a simple diagram that is also not in some way misleading. We have done our best to clearly describe the philosophy behind our approach, its implementation, and to give examples of how key processes are treated in the text.

*Section 3.1.2. P8 Lines 11 to 12 – Are these manual reactions separate to the automatically determined ones and could they be separately flagged in the supplementary material?*

We have added a short sentence here mentioning that these placeholder reactions can be found in the supplement.

*Section 3.1.2 P9 Lines 1 to 2 – How much does letting the Ox tag be inherited from NO2 sources impact on the tagging scheme (related to point 2 above).*

As pointed out by Referee #1, it would be more appropriate for the $NO_3$ (referred to implicitly here by Referee #2) produced in the reaction between $NO_2$ and $O_3$ to inherit half of its tagged identity from $NO_2$ and the other half from $O_3$. We propose to implement this change in a future version of the tagging system. Since this reaction is most likely to be important at night time (due to the longer lifetime of $NO_2$), and ozone tropospheric ozone photochemistry is more active at daytime, we do not expect this to make a large difference.

*Section 4. P12 Line 21 – 'win' should be 'in'*

This typo has been fixed.

*Section 4. P12 Line 21 and 24– Final year of the simulation is mentioned whereas it would be nice to state actual year of the simulation in which results can be obtained (i.e. 4th year).*

We have added text clarifying that we used the second year of a $NO_x$-tagged run and the third year of a VOC-tagged run for our analysis.

*Section 4. P12 Lines 23 to 24 – Is there a reason that using the tagged chemical mechanism generates different results to the original mechanism, particularly in the free troposphere?*

We would like to make it clear that the ozone simulated with our tagged model is identical to the ozone simulated by an unmodified version of the model. We have added a sentence clarifying this to the text referred to by the referee. The "different results" to which the referee refers are the differences between the actual modelled ozone and the sum of the tagged ozone tracers. We do not necessarily expect that these will be identical, due to the propagation of small numerical errors, particularly due to the advection scheme, where the spatial gradients of the tagged tracers may differ substantially from the spatial gradient of the actual modelled tracers. It would be possible to implement a "mass fixer" to remove these differences at every time step, but we believe that doing so would open up the possibility of masking real errors in the tagging scheme, so we prefer to verify that the differences between tagged species and the corresponding actual species in the model are small.

*Section 5. P12 Line 30 – I don't think that the gradient really reverses that much for ozone from anthropogenic sources, changes are more subtle.*

We are referring to the total surface ozone at this point in the text. We have made this more explicit in the revised manuscript.

*Section 5. P14. Lines 2 to 4 – Is this sentence talking about O3 only from the VOC tagging? I think it this needs to be clarified in the sentence.*

This sentence combines results from both the $NO_x$ tagging and the VOC tagging. We have added text making this explicit.

*Section 5. P15 Line 1 – I found the colours on these Figures quite hard to determine actual concentrations from. The blues seem to cover the range of between 8 and 20 ppb making it hard to identify precisely the contribution from methane. Would using a different colour scale (or different increments) provide better results?*

We have chosen a different colour scale for all of these figures. This new scale makes the transitions between contour intervals easier to determine.

*Section 5. P 15 Line 3 – Is the influence of the stratosphere stronger in winter, looks like just a shift in hemispheres.*

This comment appears to relate to point 4 in the General Comments from this referee, which relates to the ambiguity of terms like "winter" when the hemisphere is not made explicit. We have amended this part of the text to point out that stratospheric ozone makes a stronger contribution in winter in both hemispheres.

*Section 5. P 15 Line 4 – August is mentioned but no results shown to verify (See point 5 above).*

This point has been addressed in our response to point 5 in the General Comments from this referee.

*Section 5. P 16 Lines 6 to 9 – In one sentence mention is made of an upper bound on the stratospheric contribution whilst later on it is referred to as a lower bound. Please could the author clarify if this is correct.*

We are actually referring to two different methods of determining the stratospheric influence here: the use of a stratospheric tracer such as that employed by Roelofs and Lelieveld (1997), and the method of calculating the residual ozone as employed by Emmons et al. (2012). We have restructured this paragraph to make it clearer that we are contrasting two different methods.

*Section 5. P 19 Line 17 to 18 – Could you provide numbers to verify that the stratospheric contribution is lower in the VOC tagging than for NOx tagging.*

We have added two new tables (Tables 1 and 2) to the manuscript in response to the Short Comment by Colette Heald. These tables contain numbers quantifying the contribution of all sources to the tropospheric ozone burden in both the $NO_x$-tagged and VOC-tagged runs, including the stratospheric contribution. We confirm from these Tables that an extra 8.8 Tg of $O_3$ is attributed to the stratosphere when $NO_x$ is tagged, and have restructured the discussion in this section accordingly. While making this change, we realised that we had mistakenly attributed the extra stratospheric contribution to tropospheric ozone as being due to the *photolysis* of $N_2O$, when in fact it is due to the reaction of $N_2O$ with excited oxygen atoms (primarily in the stratosphere). The mistake was purely in the text of the manuscript; the process was correctly tagged in our model runs. None of the reviewers noticed this mistake, but we have nevertheless corrected this in all relevant parts of the manuscript.

*Section 5. P19 Lines 25 to 29 – Mention is made here of the limitations in the stratospheric chemistry within the model. I think it would be useful to briefly mention if anticipated future improvements to stratospheric chemistry are likely to increase or reduce the stratospheric contribution to tropospheric ozone.*

It is hard to know how to respond to this comment. Which anticipated future improvements to stratospheric chemistry is the referee referring to? In the text to which the referee refers, we merely state that a more explicit treatment of stratospheric chemistry should be used in order to be able to draw firmer conclusions about the stratosphere. We see no need to change this text in the revised version of the manuscript.

**Response to Colette Heald**

*This is a great paper and I am happy to see more work on tagging techniques at the global scale which I believe could really help us understand tropospheric ozone budgets and model diversity.*

*The initial work presented with this technique is very interesting. I wonder if the authors would consider adding a table with the quantitative contributions of the different tagged tracers to the ozone burden? These would be useful numbers to have in the literature.*

We thank Colette Heald for her positive assessment of our manuscript. Naturally we agree that a quantitative assessment of the contribution of different sources to the tropospheric ozone budget would be of wide interest. We plan several future studies using our new technique to provide exactly this, and we have made the basic tools available to the community so that other groups may also perform such studies.

But since it was relatively straightforward, we have added two new tables to the present manuscript which quantify the contributions of the sources we have chosen to tag in this study. Tables 1 and 2 contain numbers quantifying the contribution of all sources to the tropospheric ozone burden in both the $NO_x$-tagged and VOC-tagged runs. We have also disaggregated the effect of emitted CO from the anthropogenic, biogenic, and biomass burning sectors, and presented this separately for our VOC-tagged runs. This change has also flowed through into Figures 2, 4, and 6 (as well as the new figures added to the supplement).

We added these new tables at the beginning of our results section, along with some discussion which was mostly adapted from discussion of Figures 1-4 that was already present in the manuscript. This necessitated some reorganisation of our results section. Discussion of the quantitative differences between the stratospheric contributions to tropospheric ozone under $NO_x$- and VOC-tagging was added, also in response to a comment from referee #2.

[revised manuscript text omitted]